# Using machine learning methods to predict all-cause somatic hospitalizations in adults: A systematic review

Mohsen Askar[1]*, Masoud Tafavvoghi[2], Lars Småbrekke[1], Lars Ailo Bongo[2], Kristian Svendsen[1]

1 Faculty of Health Sciences, Department of Pharmacy, UiT-The Arctic University of Norway, Tromsø, Norway, 2 Faculty of Science and Technology, Department of Computer Science, UiT-The Arctic University of Norway, Tromsø, Norway

* mohsen.g.askar@uit.no

## Abstract

### Aim

In this review, we investigated how Machine Learning (ML) was utilized to predict all-cause somatic hospital admissions and readmissions in adults.

### Methods

We searched eight databases (PubMed, Embase, Web of Science, CINAHL, ProQuest, OpenGrey, WorldCat, and MedNar) from their inception date to October 2023, and included records that predicted all-cause somatic hospital admissions and readmissions of adults using ML methodology. We used the CHARMS checklist for data extraction, PROBAST for bias and applicability assessment, and TRIPOD for reporting quality.

### Results

We screened 7,543 studies of which 163 full-text records were read and 116 met the review inclusion criteria. Among these, 45 predicted admission, 70 predicted readmission, and one study predicted both. There was a substantial variety in the types of datasets, algorithms, features, data preprocessing steps, evaluation, and validation methods. The most used types of features were demographics, diagnoses, vital signs, and laboratory tests. Area Under the ROC curve (AUC) was the most used evaluation metric. Models trained using boosting tree-based algorithms often performed better compared to others. ML algorithms commonly outperformed traditional regression techniques. Sixteen studies used Natural language processing (NLP) of clinical notes for prediction, all studies yielded good results. The overall adherence to reporting quality was poor in the review studies. Only five percent of models were implemented in clinical practice. The most frequently inadequately addressed methodological aspects were: providing model interpretations on the individual patient level, full code availability, performing external validation, calibrating models, and handling class imbalance.

**Data Availability Statement:** This is a systematic review. The extracted data is available in the supplementary material.

**Funding:** The publication charges for this article have been funded by a grant from the publication fund of UiT The Arctic University of Norway. The funders had no role in study design, data collection and analysis, decision to publish, or preparation of the manuscript.

**Competing interests:** The authors have declared that no competing interests exist.

## Conclusion

This review has identified considerable concerns regarding methodological issues and reporting quality in studies investigating ML to predict hospitalizations. To ensure the acceptability of these models in clinical settings, it is crucial to improve the quality of future studies.

## Introduction

Unplanned hospital admissions and readmissions (hospitalizations) account for a significant share of global healthcare expenditures [1,2]. Interestingly, up to 35% of these hospitalizations are potentially avoidable [3]. One approach to address avoidable hospitalizations is to implement statistical and mathematical models on healthcare datasets in order to predict future hospitalization [4,5].

Previous attempts were mainly based on regression models and specific risk indexes (scores). Systematic reviews have concluded that most models had poor, inconsistent performance, and limited applicability. They also found that models utilizing health records data performed better than models using self-report data [4,6,7].

More recently, prediction models that utilize Machine Learning (ML) [8,9] algorithms have become more popular. Recent reviews emphasized the growing importance and effectiveness of ML models in predicting clinical outcomes such as hospital readmissions. These reviews concluded that ML techniques can improve readmission prediction ability over traditional statistical models. This improvement could be explained by ML models offering several advantages over traditional regression models such as flexibility, the ability to handle large, complex, high dimensional datasets, and identifying non-linear relationships [10]. The reviews also highlighted the critical role of selecting features and addressed some challenges such as transparency, the difficulty of ML models' interpretation, and the importance of handling class imbalance to enhance the models' performance. Moreover, they highlighted the importance of demonstrating the clinical usefulness of the models in practice [11–13]. A systematic analysis of readmission prediction literature proposed a comprehensive framework for ML model development detailing steps from data preparation and preprocessing to suggesting methods of feature selection and transformation, data splitting, model training, validation, and evaluation [14].

Although several reviews have considered the use of ML in predicting hospitalizations for specific diseases and conditions [15–17], none has systemically reviewed the literature on all-cause hospital admissions. We aim with this review to (i) summarize the characteristics of ML studies used in predicting all-cause somatic admissions and readmissions; (ii) provide a picture of the ML pipeline steps including, data preprocessing, feature selection, model evaluation, validation, calibration, and explanation; (iii) assessing the risk of bias, applicability and reporting completeness of the studies; and finally (iv) to comment on the challenges facing implementation of ML models in clinical practice.

## Materials and methods

The protocol of this systematic review was registered in the International prospective register of systematic reviews PROSPERO (CRD42021276721). The PRISMA, and PRISMA-Abstract guidelines [18] were followed in reporting this review, see S1 File: Section 1.

## Inclusions/Exclusion criteria

To formulate the research question, we used the PICOTS checklist [19,20]. Studies that only included non-adults were excluded. Hospitalizations were defined as all-cause somatic admissions or readmissions from outside hospitals, hence psychological-related, disease-specific, and internal admissions between wards are excluded. Emergency Department (ED) were considered portals, thus the admissions from an ED to the hospitals were included, but ED admissions followed by discharge were excluded.

Our focus is the studies performed in the ML context (both in developing steps of the model, e.g., feature engineering, or making the final predictions), so studies that only used statistical learning or risk indexes for prediction were excluded. All performance measures were reported for competing models. This review is mainly descriptive of how ML was used in predicting hospitalization; hence we chose to include studies conducted using real-world data with hospital admissions and readmissions as a valid outcome regardless of the timing of the outcome. Table 1 represents the overall inclusion criteria. A detailed description of the inclusions and exclusions is provided in S1 File: Section 2.

## Search strategy

We searched four main databases: PubMed, Embase (via Ovid), Web of Science, and CINAHL (via EBSCO) from inception dates to October 13th,2023. The search strategy was developed through the piloting of some relevant studies. Search terms were used if included in the database (MeSH for PubMed and CINAHL, and Emtree for EMBASE). We also searched 4 other databases: ProQuest, OpenGrey, WorldCat (OCLC FirstSearch), and MedNar for grey literature.

Four main search blocks were used to identify relevant studies: prediction, hospitalization, machine learning, and exclusions. The exclusion of irrelevant search words was developed by iteration and preliminary title/abstract piloting. The Boolean operators AND, OR, and NOT were used alongside truncation operators and phrase-searching. Search syntax was adapted for each database using the Polyglot tool [21] with manual supervision. The complete search syntax can be found in S1 File: Section 3.

Duplicate studies were removed using Mendeley Reference Manager (version 1.19.8, Elsevier). In cases where the reference manager was uncertain, we manually checked and removed any duplicates. Titles and abstracts were screened by two independent investigators (MA and KS), and full-text papers were retrieved for all candidate studies. The full-text screening was separately performed by MA, MT, LS, and KS. A manual search was conducted using the reference lists of the included studies to manually extract literature that did not appear in the electronic search. A list of all full-text screened studies, including those that were included and excluded with the reason(s) for exclusion, is attached to S2 File, sheet: Included & excluded

**Table 1. The criteria for study inclusion.**

| Inclusion criteria |
| --- |
| 1. Adult population |
| 2. All-cause somatic-related hospital admissions and readmissions from outside hospitals with no specified prior time |
| 3. Utilizing machine learning methods in developing or conducting prediction models |
| 4. Real-world datasets |
| 5. English language literature |

studies. The final included studies were decided by discussion between MA, KS, and LS. The descriptive results were synthesized using Pivot tables in Microsoft Excel.

### Data extraction

The data was extracted using the Critical Appraisal and Data Extraction for Systematic Reviews of Prediction Modelling Studies: The (CHARMS) Checklist [19] separately by MA, MT, and LS. For further analysis, features were grouped into administrative and clinical feature groups. The included records, extracted data, models' features, and feature groupings can be found in S2 File, sheets: CHARMS, and Features.

### Assessment of bias and applicability

Despite that the main purpose of the review is descriptive, we assessed the risk of bias and applicability using the Prediction model Risk of Bias Assessment Tool (PROBAST) [22] by MA and MT. PROBAST is a commonly used tool to assess prediction models. The tool evaluates four domains; namely: Participants, Predictors, Outcome, and Analysis. For each domain, there is a set of questions to help judge the risk of both bias and applicability concerns. If any domain was not rated "low", the overall risk of bias was considered "high". Abstracts were not assessed due to their limited information. The assessment is attached to S2 File, sheet: PROBAST.

### Quality of reporting

To assess the quality of reporting, we utilized the Transparent reporting of a multivariable prediction model for individual prognosis or diagnosis (TRIPOD) checklist [23]. We followed the suggested methodology in Constanza et al. [24] to evaluate the adherence to TRIPOD per article and per item in the reporting checklist. Each item is scored by (1 = reported, 0 = not reported, 0,5 = incomplete reporting, or not applicable = '_'). Abstracts and conference proceedings were not evaluated. We then calculated the adherence per TRIPOD item by dividing the sum of the items of all studies by the total number of studies. Adherence for each article was calculated as the sum of all TRIPOD items over the sum of all items if the reporting was complete, S2 File, sheet: TRIPOD. All abbreviations mentioned in the study are included in S2 File, sheet: Abbreviations.

## Results

Of the 7,543 records reviewed, 147 records were eligible for full-text screening. We included 16 other records by manual searching of the references. In total, 163 studies were fully screened, and 116 studies were included in the review, of which 87 peer-reviewed articles (76%), 17 conference articles, 9 abstracts, and three theses (Fig 1).

### Data extraction results

**Characteristics of the included studies.** Sixty-one studies (53%) were conducted using data from the USA, followed by Australia (seven studies), Taiwan (four studies), and Canada and Singapore, each with three studies. The oldest article is from 2005, and 2019 had the greatest number of articles (22 articles, 19%), followed by 2020 (17 articles, 15%), see Fig 2.

**Population characteristics.** Only 23 studies (20%) had a complete reporting of sample size (number of unique patients and the total number of admissions). Six studies (5%) neither reported the number of patients nor admissions (among them were 4 abstracts). The sample size varied from 371 to 4,637,294. Regarding age, 49 studies (42%) did not report an age range

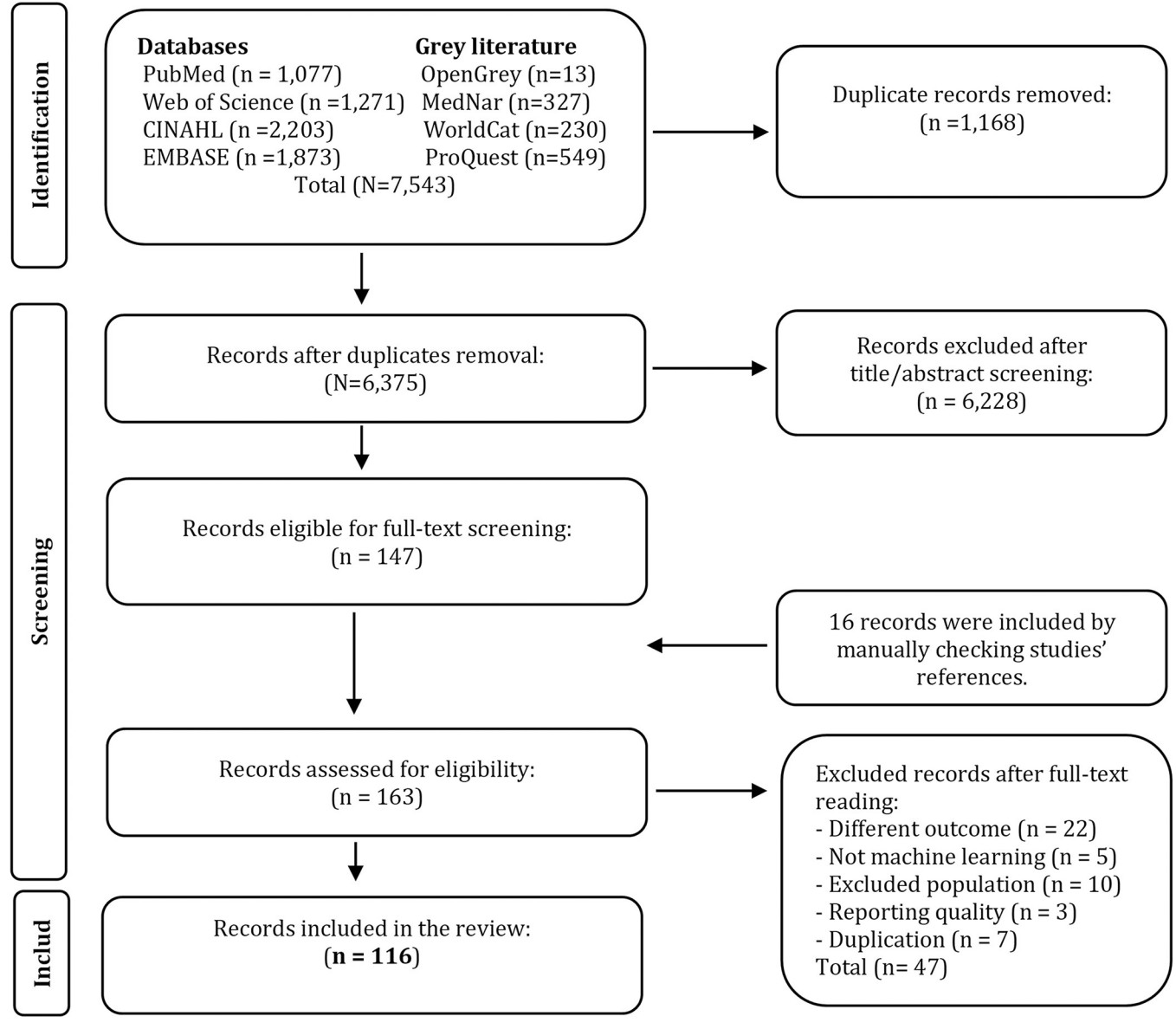

**Fig 1. A PRISMA flowchart of studies inclusion.**

of the included patients. The rest of the studies had different minimum age requirements for the studied patients.

**Outcomes characteristics.** Readmission was the outcome in 70 studies (60%), while 45 studies (39%) had hospital admission as an outcome, and one study investigated both outcomes. The readmission outcome prediction horizon varied from 24 hours to 1 year. The most frequent predicted horizon was 30-day readmission (51 studies (73% of the readmission studies), 7 other studies combined it with other readmission horizons, in total 58 studies (83%)). The dataset's inclusion period varied from 1 month to 30 years (median: 1.25 years, mean 3.2 years). Excluding rebalanced datasets, the readmission proportion varied from 0.7% to 34.6% (median 12.4%), while the admission proportion varied from 0.38% to 41% (median 17.2%).

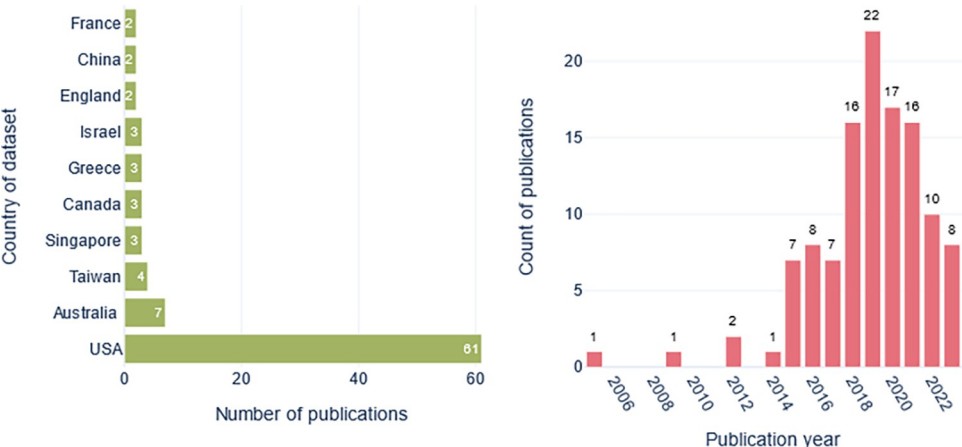

**Fig 2.** Left panel, a bar plot of the top 10 countries where the datasets originate and the number of publications. Right panel, a bar plot of the number of publications by year.

## Datasets

The number of studies that used administrative, clinical, or both types was close (40, 36, and 38 studies respectively), with two abstracts, having an unclear description of the type of dataset. Among the studies that reported an Area Under the ROC curve (AUC) and used these types of datasets (103 studies), the mean AUCs were 0.80, 0.78, and 0.77 with standard deviations (SD) of 0.08, 0.07, and 0.09 respectively. Six studies reported an AUC over 90%, while 81 studies reported an AUC from 70–90% and 18 studies reported an AUC ranging from 60–70%. Fig 3 shows the relationship between outcomes, dataset types, dataset sources, and the best model performance. S1 File: Section 4 includes detailed information on the data types, sources, and frequency of use with predicting either admission or readmissions.

**Types of features included in models.** The most used feature groups were demographics (92 studies, 79%), diagnoses (43 studies, 37%), vital signs (34 studies, 29%), and laboratory tests (28%). Fig 4 represents the most used feature groups in the included studies. Natural

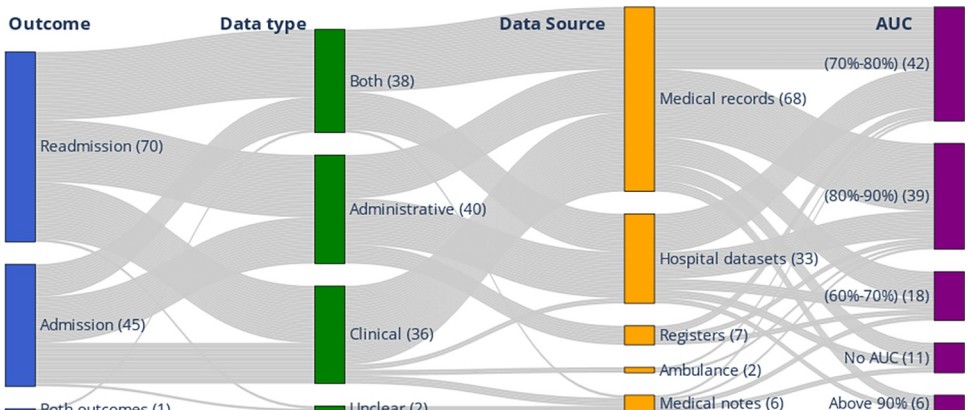

**Fig 3. Sankey diagram showing the type of outcome, datatypes and sources of datasets, and model performance by AUC.** The thickness of the streams indicates the number of records common between pairs of categories. Medical records include patient information from EHR or EMR. Hospital datasets include data from hospital information systems.

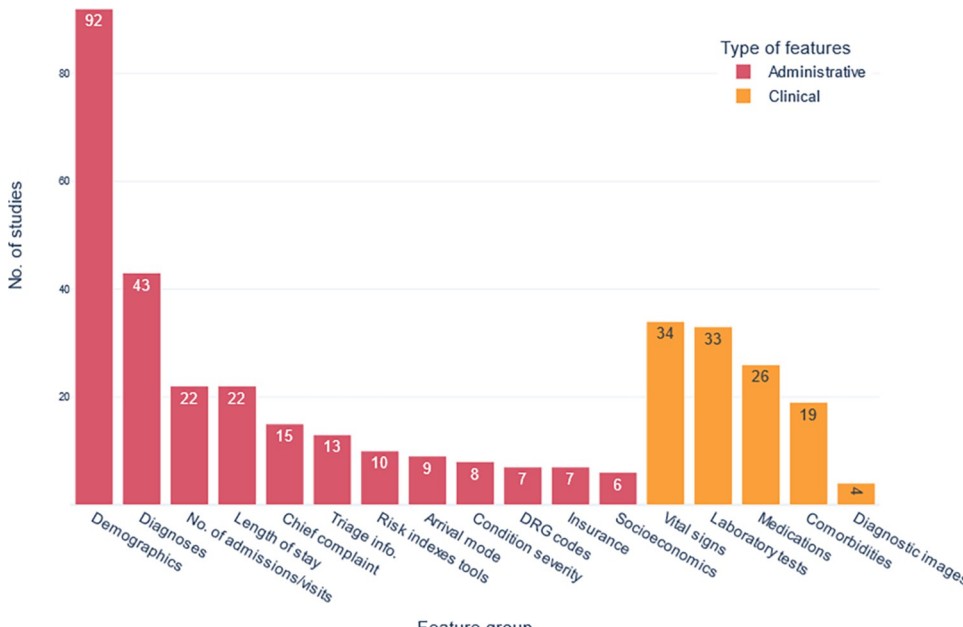

**Fig 4. The most frequently used feature groups in the retrieved studies.**

Language Processing techniques (NLP) were used in 16 studies (14%) to predict hospitalizations from clinical free-text notes.

**Missing data and data imbalance.** Missing values were not mentioned at all in 52 studies (45%). In 55 studies (47%) where the handling of how missing values were reported, the most used methods were removing the records of missing values (27 studies, 23%) and various imputation methods (25 studies, 22%) with some studies using both removing and imputation to deal with missing values.

Of 99 studies (85%) that reported class imbalance in the outcome, only 32 studies (32%) reported handling this imbalance by some technique. The most used techniques were undersampling (19 studies, 59%), oversampling (7 studies, 22%), and Synthetic Minority Oversampling TEchnique (SMOTE) (6 studies, 19%). Note that some studies tested more than one resampling method.

## Models' performance and comparison

In total, 57 different algorithms were used for predicting the outcomes. Regression models were the most frequently used algorithm group (73 studies, 63%) followed by bagging tree-based algorithms, in 61 studies (53%), and boosting tree-based algorithms in 60 studies (52%). The best-performing algorithm group was boosting algorithms in 35 studies (42%), bagging algorithms in 16 studies (19%), followed by regression and Neural Networks (NN) models in 14 studies (17%) each, see Table 2.

**Comparing the performance of algorithms.** Eighty-three studies (72%) compared the performance of multiple algorithms. Based on the results of the best-performing algorithm groups (Table 2), we compared the performance of some of these algorithm groups. DT, Bayesian models were not involved in the comparison as they did not perform best in any of the studies. Fig 5 illustrates the performance comparison between different algorithm groups in the retrieved studies.

**Table 2. Algorithms' groups and frequency of use in the included studies.**

| Algorithm Group | Included variants | No. and % of studies * | No. and % of studies performed best in** |
|---|---|---|---|
| Regression Models | Logistic, Stepwise, Penalized, Regularized, Multivariate, Multinominal, LASSO, Cox, GLM, EBM | 73 (63%) [1] | 14 (17%) |
| Bagging algorithms | RF, weighed RF, balanced RF, Causal Forests | 61 (53%) | 16 (19%) |
| Boosting algorithms | XGBoost, AdaBoost, CatBoost, GBM, LogitBoost, LightGBM, RUSBoost, Modified weight boosting stacking algorithm, GBDT | 60 (52%) | 35 (42%) |
| Neural Networks (NN) | NN, MLP, CNN, RNN, BPNN, DNN, MLNN, ANN, LSTM, GNN, Transformers NN | 48 (41%) | 14 (17%) |
| Decision Trees (DT) | DT, Conditional DT, CART, C 5.0, M5, Rpart, Conditional inference trees, Extremely Randomized Trees | 26 (22%) | 0 |
| Support Vector Machine (SVM) | Radial, Linear, non-linear, Poly SVM | 25 (22%) | 5 (6%) |
| Bayesian Models | NBC, Bayesian Network, BPM | 21 (18%) | 0 |
| Other Models | Fuzzy Fingerprint classifier, Fuzzy min-max neural networks, Fuzzy C-means clustering, Pattern recognition conditional Random Fields (CRF), KNN, QDA | 7 (6%) | 5 (6%) |

* Of the total studies (116)

** Only the studies that compared more than one algorithm (83 studies). The total number is higher than 83 (89 studies) as there were studies where more than one algorithm performed best. (1) In three studies, regression models were used for prediction while other ML algorithms were used in feature selection and other data preparation steps, thus there was no comparison between regression models and ML models in terms of prediction performance. Abbreviations arranged alphabetically: AdaBoost: Adaptive Boosting classifier; ANN: Artificial neural network; BPNN: Back-Propagation Neural Network; BPM: Bayes Point Machine; CART: Classification And Regression Tree; CatBoost: Category Boosting classifier; CNN: Convolutional Neural Network; DNN: Deep Neural Networks; EBM: Explainable Boosting Machine; GBM: Gradient Boosting Machine; GBDT: Gradient Boosted Decision Trees; GBDT: Gradient Boosted Decision Trees; GLM: Generalized Linear Models; GNN: Graph Neural Network; KNN: K-nearest neighbors; LASSO: Least Absolute Shrinkage and Selection Operator; LSTM: Long short-term memory; MLNN: Multi-Layer Neural Network; MLP: Multilayer Perceptron; NN: Neural Networks; QDA: Quadratic Discriminant Analysis; RF: Random Forest; RNN: Recurrent Neural Network; RUSBoost: Random Undersampling Boosting; XGBoost: Extreme Gradient Boosting. All abbreviations are included in S2 File, sheet: Abbreviations.

**Evaluation metrics.** AUC was the most used evaluation metric (105 studies) followed by precision, sensitivity, specificity, and accuracy (Fig 6). Thirty-seven studies (32%) reported only one evaluation metric such as AUC or accuracy without reporting a clinical performance metric such as sensitivity or specificity. Of the 105 studies that reported AUC, 18 studies (17%) reported AUC between (60–70%), 42 studies (40%) reported AUC between (70–80%), 39 studies (37%) reported an AUC between (80–90%), and only 6 studies (6%) reported AUC above 90% (Fig 3). The highest reported AUC in admission models was 95% and 99% in readmission models. The mean AUC reported in the studies that used administrative, clinical, or combined both was 0.80, 0.78, and 0.77 ((SD: 0.08, 0.07, and 0.09) respectively.

**Model calibration and benchmarking.** Only 28 studies (24%) calibrated their models using one of the calibration methods. Fig 6 represents the calibration methods used and the count of publications. Eighteen studies (16%) were benchmarked against one or more risk prediction indexes such as LACE [25], PARR [26], HOSPITAL [27] indexes, etc. The most used risk index in benchmarking was LACE index with nine studies, followed by PARR and HOSPITAL with two studies each. In all 18 studies, ML models outperformed predictions obtained from these risk indexes. A detailed comparison is attached to S1 File: Section 5.

**Model validation.** The majority of studies were trained and validated retrospectively (96 studies, 83%). Only 17 studies (15%) were trained retrospectively and tested prospectively, among them three studies performed a real-time validation. The study design was not clear in three studies. Fig 6 depicts the internal and external validation methods used in the studies.

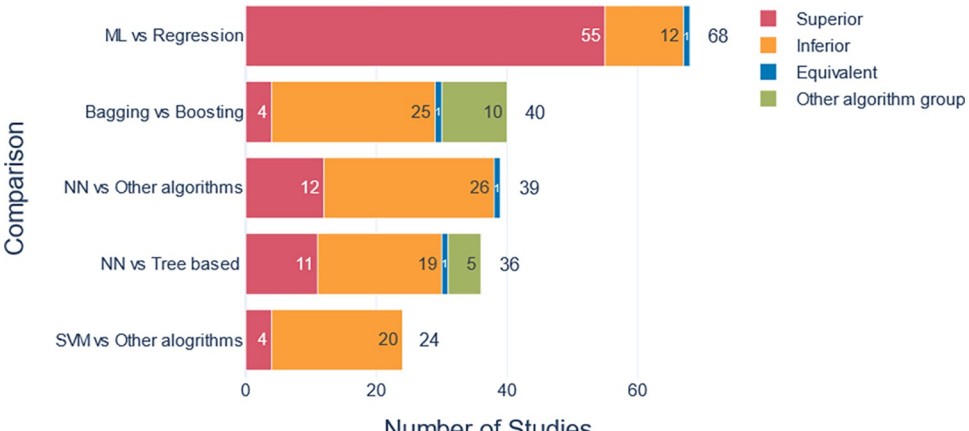

**Fig 5. A pairwise comparison of the performance of different algorithms' groups.** The numbers on each segment denote the count of publications in which the first algorithm group demonstrated superior, equivalent, or inferior performance compared to the second one. Adjacent to each bar, the total number of publications involving such comparisons was indicated.

## Model explainability and availability

Providing model interpretation on the patient level (local model interpretation) was only represented in three studies. Fig 6 represents the different interpretation methods used in the studies. Twenty studies (17%) used publicly available datasets, and 15 studies (13%) reported providing the data upon request. Only 17 studies made their code available (15%) and only six studies implemented their models in clinical practice.

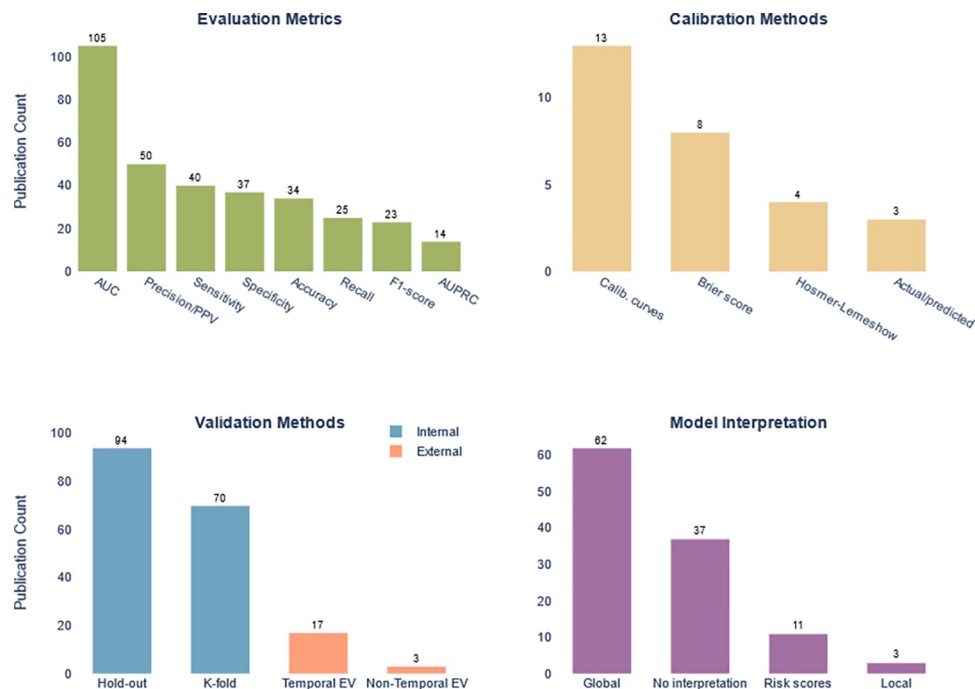

**Fig 6. Various aspects of model evaluation.** Each subplot represents the frequency of use in the reviewed studies.

## Quality of the studies

**Bias and applicability assessment.** Of 106 studies assessed, 68 (64%) were evaluated to be at high risk of bias. We evaluated 94 studies (87%) to be at low concern of applicability. Assessment results are attached to S2 File, sheet: PROBAST.

**Reporting quality assessment.** Only nine studies reported adherence to TRIPOD checklist. These studies [28–36] had generally better reporting quality (scored 17, 17, 19, 17.5, 16.5, 18, 17.5, 17, 16 out of 20) respectively. The overall median of 20 items TRIPOD adherence was 77% (IQR 63–95). The assessment of adherence to TRIPOD reveals insufficient reporting, especially in some items such as reporting the flow of participants (35% of the studies), supplementary material (52%), population characteristics (53%), reporting missing data (56%), and funding (58%) among others (Fig 7). The evaluation sheet is attached to S2 File, sheet: TRIPOD

## Discussion

To our knowledge, this is the first systematic review to focus on ML models for predicting all-cause somatic hospitalizations. Of 7,543 citations, 116 studies were included. Our review reveals the potential that ML models have in predicting all-cause somatic hospitalizations which is consistent with what is reported by both a general review of AI and machine learning and disease-specific reviews [8,9]. Our findings also raise concerns regarding the quality of the studies conducted. Therefore, despite the potential of the ML prediction framework and the superiority over traditional statistical prediction shown in many studies, there are clear issues with the quality of reporting, external validation, model calibration, and interpretation. All these aspects should support the model performance to be convenient to implement in real-

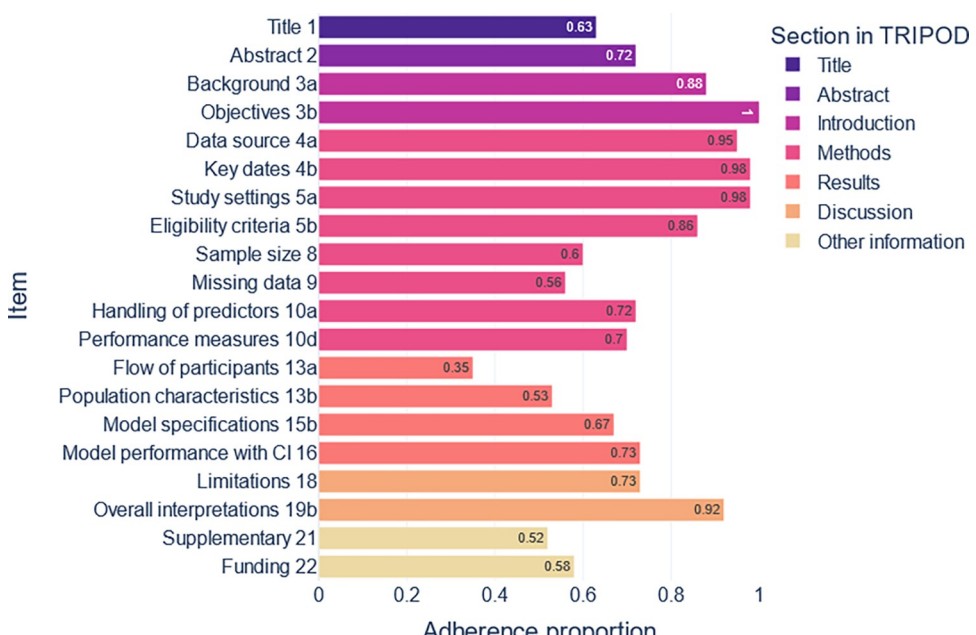

**Fig 7. Studies' adherence proportion to a range of TRIPOD checklist items.** Only explicit reporting of Confidence Intervals (CI) was considered as complete reporting. Note that all items are calculated excluding abstracts and proceedings (10 studies). Hence, some items such as missing data calculations can differ from what is reported in the result section as the calculations in the result section included all the studies.

life clinical practice. These main findings are consistent with findings from other reviews [11,12,37–39].

Most studies were based on data from the USA which can be an issue. This geographic skew limits the generalizability of the developed models, considering the differences in health-care systems and patient populations between countries [40]. As 30 day-readmission is a widely used indicator of hospital care quality [41] the majority of the included readmission studies used this indicator as an outcome.

## Datasets and features

A wide variety of data sources and types were used. We found that the performance of models trained on administrative (claims), clinical, or datasets combined both clinical and administrative variables to be close with a slight edge for models trained on administrative datasets.

The most important features varied between the different studies. This lack of convergence of risk factors is due to: i) different definitions of admissions and readmissions as an outcome for these studies, ii) the use of different feature selection methods [42], iii) the diversity of recorded features in different healthcare databases, iv) lack of standard handling of data pre-processing steps and variance of methods handling and generating variables, v) the variety in population and subpopulations, exclusion criteria, and finally, vi) the use of different risk scores and indexes which include different sets of features. This is consistent with what previous studies concluded about the difficulty of finding universal features for predicting hospitalization [43–45]. While defining general risk factors is particularly difficult for studies of all-cause hospitalizations, it may be appropriate in subpopulations (i.e., patients of specific diseases) that have more similarities and less diversity. Yet, some groups of risk factors are shown to be more common than others (Fig 4).

The most used feature groups were demographics, diagnoses, physiological measurements, and laboratory tests, respectively (Fig 4). Some studies used only one or a limited number of feature groups [46–53]. All these studies yielded generally good predictive performance suggesting that the sole use of one or limited categories of features can be enough to predict hospitalization. However, this needs to be further investigated by comparing the performance of these models that were exclusively built on one or a few feature groups with models built on several feature groups.

Some studies used Natural Language Processing (NLP) techniques to extract information from the clinical text and either combined them with other structured features [54–57] or as a sole source of data [47,52,53,58–62]. Some studies reported better prediction performance using textual data over numerical data (e.g., laboratory tests and vital signs) suggesting the existence of relevant expert knowledge within these reports [53]. We noticed an increase in applying NLP techniques in recent studies suggesting that utilizing textual data is a promising future direction for predicting hospitalizations. Incorporating NLP techniques in prediction models will provide models with a rich source of clinical information that could not be present in the tabular format of patient records. It will also improve the research scalability by the automatic extraction of relevant information rather than manual processing. Furthermore, it can provide real-time assistance for clinicians. However, some challenges should be considered such as the limited availability of sharing large, annotated datasets which are necessary for developing efficient NLP models, the current popular evaluation methods may not be clinically relevant, and the lack of transparent protocols to ensure NLP methods are reproducible [63,64].

## Data preprocessing

Elaborating on how the individual patient multiple admissions were handled in data preparation should be reported. Only 23 studies reported both a unique number of patients and the

total number of rows indicating poor reporting of this item. Reporting both the number of patients and admissions and the methods of handling multiple admissions for the same patient is important since neglecting the correlation between admissions may lead to unreliable predictions.

Similarly, less than half of the studies (47%) reported a method of handling missing values. Only a few studies (32 studies) reported handling class imbalance in the dataset. Class imbalance means that the outcome contains more samples from one class (majority class) over the other classes (minority class) [65] and represents one of the most common issues in training ML models in predicting hospitalizations. However, it is not usually taken into consideration in the readmission risk prediction literature [66]. The problem with class imbalance is that the models could be biased towards the majority class leading to a misleadingly high prediction performance [67]. Resampling techniques, especially undersampling were the most used approach. Resampling techniques involve balancing the distribution of outcome classes either by oversampling or undersampling. Oversampling involves increasing the number of instances of minority class (e.g. SMOTE), while undersampling involves randomly reducing the number of majority class instances, thus balancing the class distribution [68]. It should be noted that resampling techniques have some drawbacks such as overfitting or losing useful information which can introduce problematic consequences and hinder model learning [69,70].

## Models' performance comparisons

Evaluating model performance in health-related outcomes should be reported on two levels: model performance (e.g., AUC, F1-score, etc.) and clinical performance metrics (e.g., sensitivity, specificity, PPV, NPV, etc.) [71]. More than one-third of the studies only reported a model performance metric, with AUC as the most used one, which could limit their acceptance in clinical practice.

The analysis of different algorithms' performance confirms that no algorithm constantly performs better than the other [72]. Yet, some algorithms more frequently yield better results compared to others. In this review, we found that tree-based boosting algorithms often outperformed other algorithms (Table 2 and Fig 5). Tree-based boosting algorithms such as Gradient Boosting Machine (GBM), XGBoost, and AdaBoost, are a class of ensemble learning methods that build multiple decision trees sequentially [73]. Each new decision tree corrects the errors of previous ones by giving more focus to samples that were difficult to estimate [74]. The predictions of the trees are then combined to produce the final model prediction [75]. This group of algorithms has many advantages such as training multiple models which enhances the prediction performance over training a single one, flexibility to handle different data types, capturing non-linear patterns, and being less prone to overfitting [76].

Many studies tended to compare the performance of different algorithms on the same dataset. In this concern, we suggest that conducting even more studies with a sole focus on comparing the performance of commonly used ML algorithms is not needed unless they aim to benchmark new algorithms to the existing ones. We propose that researchers should focus on how to incorporate efforts to generalize ML models and implement them in clinical practice instead.

There is a discussion about whether ML models can offer better predictive abilities than conventional statistical models such as logistic regression (LR). While some studies found that ML models outperform regression models [11,37,77–81], others suggest that using the ML models gives no better prediction than LR [55,82,83]. In our analysis, ML models mostly performed better than regression models. This is consistent with a meta-analysis that concluded the same by comparing LR to advanced ML algorithms such as NN [84]. Regression models

performed better only in 17% of studies compared between regression and ML algorithms (Table 2 and Fig 5). This can be justified by LR being a parametric algorithm and lacking enough flexibility compared to non-parametric ones [85] or that LR has restricted assumptions which also gives favor to the less-restricted or no-assumptions algorithms [86].

We also found that ML models outperform risk indexes in prediction performance. This is reasonable because risk indexes usually contain few predictors and aim mainly for simplification of predictions, while ML models utilize more predictors and complex methods to understand the pattern in datasets. It could also be argued that ML models are developed and tested on the same dataset and may be more skilled to predict the outcome from this specific dataset and even could be overfitted for it. While risk indexes are usually developed in a setting and then validated in different datasets and settings which would mean that ML models will outperform these indexes anyway.

Finally, two studies compared ML models to clinicians' predictions. They concluded that the models outperform ED nurses in predicting admission to ED and that combining ML models with clinical insight improves the model's performance [87,88].

## Model validation

External validation (EV) should ideally be conducted on unrelated and structurally different datasets from the dataset used for model training [89,90]. If the validation dataset differs only temporally but still originates in the same settings, place, etc., it is called temporal EV and is regarded as an approach that lies midway between internal and external validation [91]. This is because the overall patients' characteristics are similar between the two datasets [92]. Our analysis shows that there was a clear shortage in terms of EV of models. Most of the EV performed can be regarded as temporal EV. Although recent studies indicate an increased awareness of EV, maintaining EV continues to be a critical step in the current development of ML models [93]. However, there are still several obstacles facing ML models' generalizability. These obstacles can be categorized as either model-related or data-related. Model-related obstacles include issues with transparency in model development and results reporting. Data-related ones include the diversity of data structure, formats, population, etc. across different healthcare systems, and the lack of a standardized data preprocessing framework. Additionally, the strict health data privacy regulations.

Adopting Common Data Models (CDMs) [94], designing a comprehensive and widely accepted framework for data preprocessing, and implementing Federated Learning (FL) [95] could help address these issues. In S1 File: Section 6, we provide a more detailed explanation of these obstacles and solutions.

## Model explainability and availability

Model interpretation is of great importance in predicting health-related outcomes. Global model interpretation involves describing the most important rules and most influential features that the model learned in the training steps [96], while, local model interpretation refers to explaining how the model derived each individual prediction (i.e., for each patient) [97,98]

In our analysis, 62 studies (53%) introduced global interpretation to their model in the form of feature importance, or a risk score, for example [28,99,100], while only three studies presented methods for local interpretation [56,78]. Introducing both global and local model interpretation is important to increase the trustworthiness and to enhance the implementation of these models in practice [101–104].

Few studies made their dataset (20 studies) or code (17 studies) publicly available. To facilitate the technical reproducibility of the model, publishing both datasets and code is necessary.

Indeed, healthcare datasets contain patients' confidential information which hinders publishing them. Hence, some suggestions were reported to partially solve this issue such as publishing a simulated dataset [105], providing complementary empirical results on an open-source benchmark dataset [106], or sharing model prediction and data labels to allow further statistical analysis [107]. There is also no doubt that publicly available datasets such as MIMIC-III [108], have boosted ML research and opened many chances to develop ML in the health domain. MIMIC-III has been cited more than 3,000 times to date. The dataset has enabled numerous studies that focus on developing predictive models and enhancing clinical decision support systems [109,110].

Reporting model developing codes and performed experiments can help understand the final methodology, accelerate the overall development, and ensure that models are safeguarded from data leakage and other downfalls in model development [71,111]. Additionally, reporting the software and package versions is also necessary. Many decisions taken by algorithms are taken silently through the default setting of the different packages leading to differences in results when the experiment is repeated even on the same dataset [112].

## Bias risk and applicability

More than 60% of the assessed studies had a high risk of bias in line with other reviews' findings [38,39,113]. Twelve studies were found to have a high concern of applicability. However, factors such as variability of populations, settings, and dataset characteristics are anticipated to further constrain the applicability of these studies.

In general, we observed poor quality of reporting in the studies. This is consistent with findings in other studies [24,114,115]. Poor reporting quality raises concerns about the reproducibility of models [105]. Studies that adhered to TRIPOD had better scores than those that didn't. This points to the importance of adherence to a reporting checklist in ML studies, especially in the health domain. It also raises the need to develop ML-specialized checklists for quality assessment and reporting quality. Ongoing research is currently addressing this requirement [116]. In S1 File: Section 7, we suggest a reporting scheme for ML studies.

## Limitations

We identified the relevant literature from eight databases, but we have not approached authors for missing information on the studies. This is due to the considerable amount of missing information which could have impacted the assessment of bias risk. Our results are also limited by the fact that most of the reviewed studies were based on data from the USA which limits the generalizability because of the differences between populations and healthcare systems between countries. To address this limitation, future studies should aim to include diverse datasets from various countries and healthcare settings. Additionally, more efforts should be directed to compare models from different populations and settings to understand their limitations in different contexts. Assessing the quality of studies was also limited by not being able to access their code scripts. Potential publication bias also limits the ability of the review to comprehensively evaluate the overall results. Additionally, the variability of reporting varied significantly between the studies which can affect the reliability of the findings.

The heterogeneity and differences in healthcare systems and patient populations across countries and ML algorithms and settings limit the comparisons of results between studies and make it more difficult to harmonize the results of different models. Due to this heterogenicity, we had to make decisions regarding the inclusion criteria which may have caused us to miss relevant studies. Finally, only literature published in English was included which also limits our insights to the overall picture of ML development globally.

## Conclusions

The main purpose of the review was to describe how ML was used in predicting all-cause somatic hospitalizations. The review raises some concerns about the quality of data preprocessing, the reporting quality, reproducibility, local interpretation, and the external validity of many studies. The quality of studies needs to improve to meet the expectations of clinicians and stakeholders before using these models in clinical practice. We recommend that future studies should prioritize generalizing ML models and integrating them into clinical practice.

## Supporting information

**S1 File. Includes: Section 1: PRISMA checklist, Section 2: Detailed inclusion/exclusion criteria, Section 3: Literature search syntax, Section 4: Studies data sources, Section 5: Benchmarking with risk indexes, Section 6: A comment on the generalizability of ML models, and Section 7: A suggestion of reporting checklist specifically for ML models in structured datasets.**
(DOCX)

**S2 File. Includes: Sheet: Abbreviations, (Sheet: CHARMS) include the extracted data for review studies and studies citations, (Sheet: Features) includes feature-related extractions, (Sheet: PROBAST) includes the applicability and risk of bias assessment, (Sheet: TRIPOD) includes the reporting quality assessment, and (Sheet: Included & excluded studies) includes full-text screened studies with the reason(s) of exclusion.**
(XLSX)

## Author Contributions

**Conceptualization:** Mohsen Askar, Kristian Svendsen.

**Data curation:** Mohsen Askar, Masoud Tafavvoghi, Lars Småbrekke, Kristian Svendsen.

**Formal analysis:** Mohsen Askar, Kristian Svendsen.

**Investigation:** Mohsen Askar, Masoud Tafavvoghi, Lars Småbrekke, Kristian Svendsen.

**Methodology:** Mohsen Askar, Lars Småbrekke, Kristian Svendsen.

**Project administration:** Kristian Svendsen.

**Resources:** Mohsen Askar.

**Software:** Mohsen Askar.

**Supervision:** Lars Småbrekke, Lars Ailo Bongo, Kristian Svendsen.

**Validation:** Mohsen Askar, Masoud Tafavvoghi.

**Visualization:** Mohsen Askar.

**Writing – original draft:** Mohsen Askar.

**Writing – review & editing:** Mohsen Askar, Masoud Tafavvoghi, Lars Småbrekke, Lars Ailo Bongo, Kristian Svendsen.

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
