## [Decision Letter · Decision Letter 0]

29 Apr 2024

PONE-D-24-04075Using machine learning methods to predict all-cause somatic hospitalizations in adults: A systematic review.PLOS ONE

Dear Dr. Askar,

Thank you for submitting your manuscript to PLOS ONE. After careful consideration, we feel that it has merit but does not fully meet PLOS ONE’s publication criteria as it currently stands. Therefore, we invite you to submit a revised version of the manuscript that addresses the points raised during the review process.

We look forward to receiving your revised manuscript.

Kind regards,

Tariq Jamal Siddiqi

Academic Editor

PLOS ONE

Journal Requirements:

The publication charges for this article have been funded by a grant from the publication fund of UiT The Arctic University of Norway.

Reviewers' comments:

Reviewer's Responses to Questions

**Comments to the Author**

1. Is the manuscript technically sound, and do the data support the conclusions?

Reviewer #1: Yes

2. Has the statistical analysis been performed appropriately and rigorously? 

Reviewer #1: Yes

3. Have the authors made all data underlying the findings in their manuscript fully available?

Reviewer #1: Yes

4. Is the manuscript presented in an intelligible fashion and written in standard English?

Reviewer #1: Yes

5. Review Comments to the Author

**Reviewer #1**: 1. While the introduction mentions the shift towards ML techniques, providing a brief rationale for why ML is superior to traditional regression models and specific risk indexes would strengthen the argument. For example, it could be helpful to highlight ML’s capacity to manage complex, high-dimensional datasets and identify nonlinear relationships.

2. The introduction cites studies that highlight the superiority of ML techniques. Providing a brief summary or key findings from these reviews would place the current study in the perspective of the existing literature and help readers understand the progression of research in the field of AI/ML.

3. In lines 53-54, authors have mentioned that recent reviews have proposed a framework for ML model development. This point needs to elaborated further.

4. In the methods section, while the authors have mentioned the removal of duplicate studies, providing details on the methodology used for duplicate removal (e.g., software/tool used) would enhance transparency of methods.

5. The authors should consider providing a brief explanation of the PROBAST tool and its domains to help readers understand how bias and applicability were assessed.

6. Line 162: In population characteristics, the authors can include total number of patients, and other baseline characteristics from the included studies.

7. Line 306-310: This can be highlighted as a limitation of this study, and can be elaborated further. Moreover, future directions are required as to how future studies can particularly tackle this.

8. Line 328, the sentence lacks a necessary comma. To improve clarity, it should be as follows: "The most used feature groups were demographics, diagnoses, physiological measurements, and laboratory tests, respectively."

9. In the "datasets and features" section of the discussion, the authors made a noteworthy observation regarding the increasing use of Natural Language Processing (NLP) techniques in recent studies. This trend shows a promising avenue with regard to utilizing textual data in predicting hospitalizations. Therefore, it is suggested to further discuss the potential benefits and challenges associated with employing NLP techniques for extracting information from clinical text.

10. In lines 340-351, the authors noted that a few studies reported handling class imbalance in the dataset and that resampling techniques, especially under sampling were the most used approach. To enhance reader comprehension, it would be beneficial to succinctly define and explain these techniques for those who may not be familiar with the terminology, thus improving overall reader understanding.

11. Similarly, in section ‘Models’ performance comparisons’, the authors mentioned a key finding that tree-based boosting algorithms often outperformed other algorithms. I suggest that the authors briefly explain what these algorithms are, how they work, significance of these algorithms and why they outperformed others.

12. In lines 399-400, the authors state that” In Supplementary 1: S6, we summarize the problems facing EV and the generalization of ML models along with some suggested solutions.” It would enhance the significance of the discussion to incorporate a brief overview of these challenges and solutions.

13. In lines 417-418, the authors emphasize the pivotal role of open-source datasets like MIMIC-III [94] in boosting machine learning (ML) research within the healthcare domain. To underscore the significance of these datasets further, the authors could cite evidence or statistics demonstrating the widespread impact of these open-source datasets. This will support the need of leveraging open data for advancing ML applications in healthcare.

6. PLOS authors have the option to publish the peer review history of their article (what does this mean?). If published, this will include your full peer review and any attached files.

Reviewer #1: No

---

## [Author Response · Author response to Decision Letter 0]

3 Jun 2024

Responses are given in the "Response to reviewers" letter.

---

## [Decision Letter · Decision Letter 1]

14 Jun 2024

PONE-D-24-04075R1Using machine learning methods to predict all-cause somatic hospitalizations in adults: A systematic review.PLOS ONE

Dear Dr. Askar,

Thank you for submitting your manuscript to PLOS ONE. After careful consideration, we feel that it has merit but does not fully meet PLOS ONE’s publication criteria as it currently stands. Therefore, we invite you to submit a revised version of the manuscript that addresses the points raised during the review process.

We look forward to receiving your revised manuscript.

Kind regards,

Tariq Jamal Siddiqi

Academic Editor

PLOS ONE

Journal Requirements:

Reviewers' comments:

Reviewer's Responses to Questions

**Comments to the Author**

1. If the authors have adequately addressed your comments raised in a previous round of review and you feel that this manuscript is now acceptable for publication, you may indicate that here to bypass the “Comments to the Author” section, enter your conflict of interest statement in the “Confidential to Editor” section, and submit your "Accept" recommendation.

Reviewer #2: All comments have been addressed

2. Is the manuscript technically sound, and do the data support the conclusions?

Reviewer #2: Yes

3. Has the statistical analysis been performed appropriately and rigorously? 

Reviewer #2: Yes

4. Have the authors made all data underlying the findings in their manuscript fully available?

Reviewer #2: Yes

5. Is the manuscript presented in an intelligible fashion and written in standard English?

Reviewer #2: Yes

6. Review Comments to the Author

**Reviewer #2: **The authors have submitted a revised study on using machine learning methods to predict all-cause somatic hospitalizations in adults.

- the authors have extensively revised and addressed all the comments provided in the first round of revision

- this study can benefit with a shorter introduction, which will improve readability of this paper.

- authors should consider mentioning company and version name for the referencing software of Mendeley

- instead of S2 or S3, authors can write Supplementary Table 3 or Supplementary Figure 2 etc etc.

- in my opinion, full forms of abbreviations mentioned in the table should be provided in a footnote or within somewhere in the title of the table, and not in S2 sheet.

- the authors should submit a document with double spaced text to improve readability for the reviewers as well

7. PLOS authors have the option to publish the peer review history of their article (what does this mean?). If published, this will include your full peer review and any attached files.

Reviewer #2: No

---

## [Author Response · Author response to Decision Letter 1]

12 Jul 2024

We have addressed the suggestions and responded to all reviewers' comments.

---

## [Editor Report · Decision Letter 2]

7 Aug 2024

Using machine learning methods to predict all-cause somatic hospitalizations in adults: A systematic review.

PONE-D-24-04075R2

Dear Dr. Askar,

We’re pleased to inform you that your manuscript has been judged scientifically suitable for publication and will be formally accepted for publication once it meets all outstanding technical requirements.

Kind regards,

Tariq Jamal Siddiqi

Academic Editor

PLOS ONE
---

## [Editor Report · Acceptance letter]

14 Aug 2024

PONE-D-24-04075R2 

PLOS ONE

Dear Dr. Askar, 

I'm pleased to inform you that your manuscript has been deemed suitable for publication in PLOS ONE. Congratulations! Your manuscript is now being handed over to our production team.

Kind regards, 

on behalf of

Dr. Tariq Jamal Siddiqi 

Academic Editor

PLOS ONE